



# Onset and propagation of drought into soil moisture and vegetation responses during the 2012-2019 drought in Southern California

Maria Magdalena Warter[1], Michael Bliss Singer[1,2,3], Mark O. Cuthbert[1,2,4], Dar Roberts[5], Kelly K. Caylor[3,5,6], Romy Sabathier[1] and John Stella[7]

[1]School of Earth and Ocean Sciences, Cardiff University, Cardiff, CF10 3AT, United Kingdom
[2]Water Research Institute, Cardiff University, Cardiff, CF10 3AT, United Kingdom
[3]Earth Research Institute, University of California Santa Barbara, Santa Barbara, CA 93106-3060, USA
[4]Connected Waters Initiative Research Centre (CWI), School of Civil and Environmental Engineering, UNSW Sydney, NSW 2052, Australia
[5]Department of Geography, University of California Santa Barbara, Santa Barbara, CA 93117, , USA
[6]Bren School of Environmental Science and Management, University of California Santa Barbara, Santa Barbara, CA 93117, USA
[7]Department of Forest and Natural Resources Management, State University of New York College of Environmental Science and Forestry, Syracuse, NY 13210, USA

*Correspondence to*: Maria Magdalena Warter (warterm@cardiff.ac.uk)

## Abstract

Despite clear signals of regional impacts of the recent severe drought in California within Central Valley groundwater storage and Sierra Nevada forests, our understanding of how this drought affected soil moisture and vegetation responses in lowland grasslands is limited. In order to better understand the resulting vulnerability of these landscapes to fire and ecosystem degradation, we aimed to generalize drought-induced changes in subsurface soil moisture and to explore its effects within grassland ecosystems of Southern California. We used a decadal in situ dataset of high-resolution climate and soil moisture from two grassland sites (coastal and inland), alongside greenness (NDVI) data from Landsat to explore drought dynamics in environments with similar precipitation but contrasting evaporative demand. Analysis of data from 2008 to 2019 showed that the negative impacts of prolonged net precipitation (netP) deficits on vegetation at the inlands site were buffered by fog and moderate temperatures at the coastal site. During the drought, the region experienced an early onset of the dry season, resulting in premature senescence of grasses by mid-April. We developed a parsimonious soil moisture balance model that captures dynamic vegetation-evapotranspiration feedbacks using netP-NDVI relationships as a leading indicator. We then analyzed the links between climate, soil moisture, and vegetation greenness over decadal timescales, exploring the impacts of plausible climate change scenarios that reflect changes to precipitation amounts, their seasonal distribution, and evaporative demand. We found that all scenarios generate early, extreme soil moisture deficits during drought below a vegetation stress threshold,



further intensifying early dry season onset and vegetation die-off. These changes suggest potential increases in the risk of wildfires in this and similar regions under climate change, as well as increased grassland ecosystem vulnerability.



## 1 Introduction

A severe drought between 2012 and 2016 affected most of the state of California (USA), resulting in substantial impacts to water resources and ecosystems (NDMC, 2020; Prugh et al., 2018; Shukla et al., 2015; Williams et al., 2015), yet current understanding of the California drought's impacts is based on research within particular regions and biomes. Consecutive years of low precipitation, above average temperatures and extremely dry conditions (meteorological drought) over this drought period resulted in severely reduced snowpack, streamflow and groundwater storage (hydrological drought), periods of

increased soil moisture deficit, and elevated vegetative stress (agricultural drought), with dramatic effects on upland forest dieback and tree mortality (Berg et al., 2017; Diffenbaugh et al., 2015; Swain et al., 2014; Williams et al., 2015). Although, the entire state experienced drought effects to some degree, there were notable differences in vegetation responses between Northern and Southern California (Dong et al., 2019). In upland forests within the Sierra Nevada Mountains, studies found large-scale canopy water loss and forest die-back, as a result of the accumulated precipitation deficits, increased evaporative

demand and soil moisture drying during the drought (Asner et al., 2016; Fettig et al., 2019; Goulden et al., 2019), whereas there were overall greater declines in vegetation greenness in Southern California (Dong et al., 2019). Little is known, however, about the impacts and propagation of drought through shallow soil moisture and the effects on vegetation in lowland areas, especially within water-limited regions where grasses and shrubs dominate the landscape. These lowland water-limited ecosystems comprise complex relationships between vegetation and water availability that affect the spatial patterns and extent

of different vegetation types, as well as the relative responses of different species to drought stress (Caylor et. al., 2006; Caylor et. al., 2009; D'Odorico et.al., 2007; Okin et. al., 2018). However, as climate change impacts to the water balance progress, there is a need to better understand how climate (temperature, precipitation) and soil water availability drive vegetation dynamics in lowland grasslands. The increasing loss of grassland ecosystems furthers the threat of overall land degradation and encroachment of invasive species, which ultimately feeds back into heightened vulnerability of these ecosystems to water

deficits in semiarid landscapes under climate change (Gremer et al., 2015; Lian et al., 2020). Here we explore the links between climate, soil moisture, and vegetation during the recent California drought and analyze the potential consequences of future climate scenarios to further our understanding of vegetation dynamics in lowland dryland grassland ecosystems.

Grassland ecosystems throughout Southern California naturally exhibit green and senescent (brown) periods each year, due to the region's strong Mediterranean climate, which makes these ecosystems prone to fire during the dry season. Although such

fires are part of the natural ecosystems of Southern California, they are also capable of encroaching on inhabited areas with disastrous effects (e.g., huge areas are currently burning due to fires spreading through grasslands in many Western states at the time of submitting this manuscript). Rising soil moisture deficits due to meteorological droughts can cause early senescence of vegetation, thus priming grasslands for intense wildfires, while also modifying species composition, runoff responses, and



nutrient dynamics (Lian et al., 2020, Ludwig et al., 2005; McDowell et al., 2008; Michaelides et al., 2009). In recent decades,
wildfire extent has increased substantially in Southern California, due to increased evaporative demand, reduced snowpack in
mountainous areas and loss of dry season precipitation making native grasslands more susceptible to non-native species
invasion and furthering the loss of sage scrub, and conversion to invasive habitats throughout the region (Singh and Meyer,
2020; Williams et al., 2019). The most destructive fires often occur at the end of the dry season when moisture content of live
and dead fuels is severely reduced after months of warm and dry weather (Keeley et al. 2016; Williams et al., 2019). In
Southern California, the spread of wildfires is further exacerbated by the Santa Ana winds, which can quickly promote fires
across large areas, threatening residential communities and forest ecosystems (Williams et al., 2019). One example are the
cascading effects of wildfire, subsequent rains, and debris flows that devastated Montecito in Santa Barbara County in 2018
(Oakley et al., 2018). Further warming trends are projected, leading to increasingly drier soil moisture conditions, increased
precipitation variability and heightened evaporative demand. Under future climate change significant changes in rainfall
intensity are expected throughout dryland areas (Singer & Michaelides, 2017; Singer et al., 2018), with drier spring and autumn
periods and an increase in subsequent dry years throughout California (Pierce et al., 2018). Such climatic conditions would
likely further increase fuel aridity and wildfire potential, and lead to a shift in future fire regimes with more frequent and
intense wildfires throughout the western US (Abatzoglou et al., 2016; Williams et al., 2019), thus potentially increasing the
overall vulnerability of grasslands and surrounding communities.

Currently, the vulnerability of California grasslands to future climate change is classified as 'moderately high', with some
studies estimating a substantial loss of grassland habitats by the end of the 21st century (Thorne et al., 2016; Wilkening et al.,
2019). The greater vulnerability of vegetation to drought in Southern California (compared to Northern California) and a
continuing trend of aridification in this region will likely pose a compounding challenge to lowland vegetation and water
resources throughout the entire US Southwest (Dong et al., 2019). Deep soil moisture drying has been shown to affect forest
vegetation in uplands (Goulden et al., 2019). Grasslands are known to be highly sensitive to variations in precipitation and
evaporative demand, which makes them particularly vulnerable to future climate change and shifts in precipitation variability
(Gremer et al., 2015; Reynier et al., 2016). Although many grass species are adapted to dry periods, a better understanding of
the responses of lowland grassland vegetation to time-varying soil moisture stress associated with precipitation variability
induced by climate change is essential to advance our knowledge and capabilities to mitigate the potential negative impacts of
drought on these ecosystems. Higher temperatures and increased evaporative demand in the future may shift soil moisture



conditions towards drier average conditions, thereby increasing the risk of extreme droughts and stronger summer heatwaves (Ault et al., 2016, Lian et al. 2020).

Advances in remote sensing has provided new, spatially explicit direct and indirect observations of vegetation dynamics and moisture availability(Coates et al., 2015; Liu et al., 2012; Small et al., 2018). Proxy estimates of soil moisture, such as the Palmer Drought Severity Index (PDSI) or Standardized Precipitation Evaporation Index (SPEI), are useful, but they do not directly focus on soil water stress and its associated impacts on ecosystems (Berg et al., 2018). While land-atmosphere models may be beneficial in addressing water and energy fluxes (Bonan, 1996; Davin et al., 2016; Rosolem et al., 2013) they are often overly complex (requiring too many unconstrained parameters) for site-based assessments of climate-driven soil moisture variations. The Food and Agriculture Organization (FAO) developed a well-established approach to estimate soil moisture for agricultural purposes (Allen et al, 1998), which has also proven to be useful for other non-agricultural applications (Cuthbert et al., 2013; Cuthbert et al., 2019). This simple soil moisture balance approach shows promise for understanding drought propagation into soil moisture. In this study we build upon the FAO approach by including dynamic interactions between vegetation and climate, through the incorporation of remotely sensed data to represent the relationship between soil moisture and vegetation and use historic soil moisture and climate data to investigate the evolution of soil moisture during the recent drought. Soil moisture is our key drought metric of interest, as it inherently links precipitation, evaporative demand and Normalized Difference Vegetation Index (NDVI). In previous studies NDVI has been found to be most strongly related to soil moisture of the concurrent month and also highly correlated to multi-month precipitation averages (Farrar et al., 1994). The timing of vegetation growth and die-off is also strongly related to seasonal fluctuations in water availability to plants, especially in annual grasslands, so the assessment of soil moisture and greenness is essential for vegetation drought monitoring (Liu et al., 2012; Small et al., 2018). In this study we leverage these relationships to identify the responses of grassland vegetation to declining moisture availability and changing precipitation variability.

Our primary objective was to understand the broader patterns in the soil moisture and vegetation responses to climate forcing and to advance the understanding of how drought propagates through shallow soil moisture and affects lowland grassland vegetation. We investigated: (i) how local soil moisture evolved over the recent California drought; (ii) how changes in precipitation amounts and timing affected soil moisture dynamics and grassland vegetation; and (iii) how soil moisture might respond to more prolonged dry periods under plausible climate scenarios. We employed NDVI from Landsat alongside long-term, high resolution meteorological and soil moisture data from two distinct grassland locations in Santa Barbara County with contrasting climate conditions due to orography and air flow affecting evaporative demand: a coastal and an inland site. We used these data to parameterize a simple parsimonious single-layer soil moisture balance model for generalizing the impact of



climate on plant available water in grassland ecosystems. We also developed a leading indicator of greenness based on net

precipitation that is used in our modeling framework to explore how plausible climate change scenarios would affect soil

moisture and vegetation for this region in the future.

## 2 Data and Methods

### 2.1 Study Sites

Soil moisture is essential for plant growth and -health and accordingly, there are strong seasonal responses of vegetation to

temperature and precipitation (Coates et al., 2015; Roberts et al., 2010). In this study we focused on two grasslands sites in

Santa Barbara County in Southern California. The natural geography of this region is characterized by coastal plains, oak

woodlands and a rugged mountain range (Roberts et al., 2010). Two sites were chosen from a network of several sites as they

had the best data availability spanning over 10 years, while also representing the diverse geography of the region: a coastal

grassland plain and an inland grassland site, north of the Santa Ynez Mountains (**Figure 1**). Both sites are characterized by a

Mediterranean climate, with strongly seasonal precipitation during the winter and prolonged dry periods in summer. The

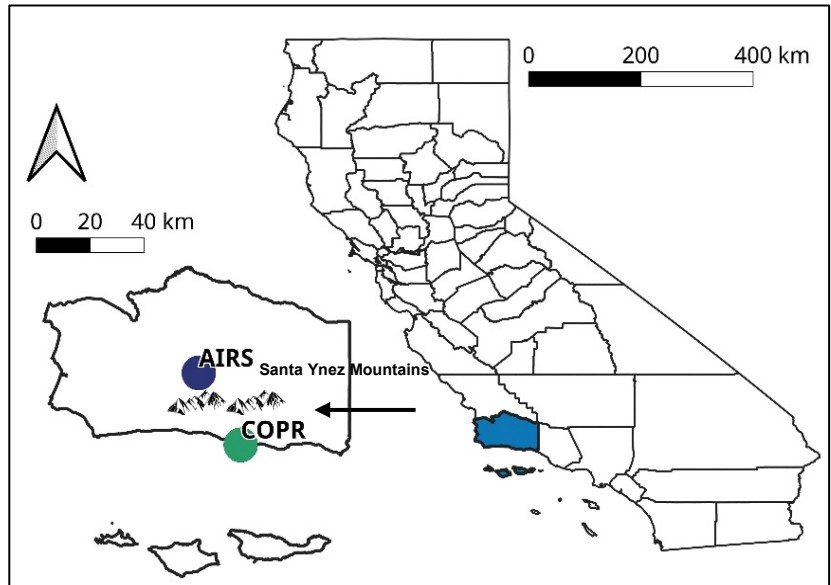

**Figure 1**

[caption] Figure 1: Location of stations in Santa Barbara County showing the coastal grassland site
(COPR, green), with a marine microclimate and the semi-arid inland grassland site (AIRS, blue) north of the
Santa Ynez mountain range.





majority of precipitation falls between November and March, with an average of 352 mm (coastal) and 314 mm (inland) per water year (October-September). Previous studies have shown that growing season water availability strongly controls annual growth cycles and senescence of vegetation at these sites (Liu et al., 2012; Roberts et al., 2010).

The coastal site is the Coal Oil Point Reserve at an elevation of 6 mASL. The dominant vegetation at this site is classified as introduced European grassland with several non-native species, including a range of annual grasses and forbs. Species vary significantly between years, due to rainfall variability, however wild oat grass (*Avena fatua*) dominates the landscape. The inland site is situated at Sedgwick Reserve Airstrip in the Santa Ynez Valley on the University of California's Sedgwick Natural Reserve at an elevation of 381 mASL. This site is an open grassland and not used for grazing. There is a higher species

variability than at the coastal site (mostly in the form of forbs) and includes several annual non-native grasses, such as various brome grasses (*Bromus hordeaceaus L., Bromus diandrus*) and wild oat. The inland site is situated in a relatively dry valley in the rain shadow of the Santa Ynez mountain range, resulting in higher evaporative demand in summer due to higher temperatures, compared to the coastal site. Temperatures are more moderate at the coastal site due to the presence of cooler, moister ocean air, coastal stratus clouds and thus lower insolation, enhanced by a coastal current, all of which reduce the

overall evaporative demand (Roberts et al., 2010). The coastal and inland sites also vary in soil textural properties and water holding capacity. Soil samples from several depths were taken at the time of sensor installation in 2007 and texture, porosity, field capacity and wilting point were determined in the lab. Soil types vary from clay loam at the coastal site to loam at the inland site, with distinctly higher sand contents at the inland site and a higher percentage of clay at the coastal site (**Table S 1**).

**2.2 Data**

We used meteorological and soil moisture data from a network of several sites where data has been continuously recorded at 15-min resolution since 2007 by the University of California Santa Barbara for educational purposes (Roberts et al., 2010). The data are publicly available and continuously updated (http://www.geog.ucsb.edu/ideas/). Meteorological data from each station includes air temperature (T), relative humidity (RH), short wave and longwave radiation, wind speed and direction, and

precipitation (P) among others. For each site, we extracted daily maximum daytime temperatures, humidity and precipitation totals and calculated monthly averages to define the meteorology of the drought. We used evaporative demand variables to calculate potential evapotranspiration (PET) through Penman Monteith. We also analyzed the date of onset (day of the year of last recorded precipitation for more than three months) and length of the dry season for each year, and compared the timing between drought and non-drought periods. Two-sample Kolmogorov-Smirnov (KS) tests and/or Pearson's correlation were

used to determine statistical differences between non-drought and drought periods and to quantify correlations between



variables, such as T, RH, P, PET, netP (P – PET losses), soil moisture saturation, and Normalized Difference Vegetation Index (NDVI). Further information on this is given in section 2.4.

Soil moisture and temperature were measured using in-situ Stevens Hydro Probes and volumetric soil water content was measured at three different depths (10, 20 and 50 cm at the coastal and 15, 23 and 46 cm at the inland site) (Roberts et al.,

2010). For the purposes of this study we are using the shallowest soil moisture (10 and 15cm) as reference. We present historical soil moisture as degree of saturation, ranging from dry (0%) to fully saturated (100%), defined as the ratio of (measured) volumetric moisture content to the volume of pore space (porosity), rather than raw volumetric water content. This enables direct comparison of soil moisture between two sites with differing soil properties and characteristics. While the data recovery for both meteorological stations was continuous for the period of interest, the soil moisture probes at the inland site

experienced significant data loss between 2016 – 2018, due to battery and sensor failure and these gaps in the data are indicated in the results (e.g. **Figure 7b**). Further detailed explanation of the experimental set-up and station equipment can be found in Roberts et  al. (2010) and at http://www.geog.ucsb.edu/ideas/.



## 2.3 Historical Climate

The United States Drought Monitor (USDM, https://droughtmonitor.unl.edu/) defines drought as a moisture deficit of such
severity that it causes social, environmental, or economic effects. It identifies and labels areas of drought within the United
States based on a semi-quantitative intensity scale, derived from a combination of key indicators and information on soil
moisture, precipitation, streamflow and drought severity, along with local condition and impact reports and ranges from D0
(Abnormally Dry) to D4 (Exceptional Drought) (NDMC, 2020). The drought affected the majority of the state of California
between 2012-2016 (e.g., Dong et al., 2019) at varying levels according to the USDM (**Figure 2a**). Based on the USDM, Santa
Barbara County was under continuous drought conditions a lot longer: from 2012-2019. The entire county was under 'extreme'
(D3) to 'exceptional' (D4) drought from mid-2013 until early 2017, with the entire area remaining in the most severe category

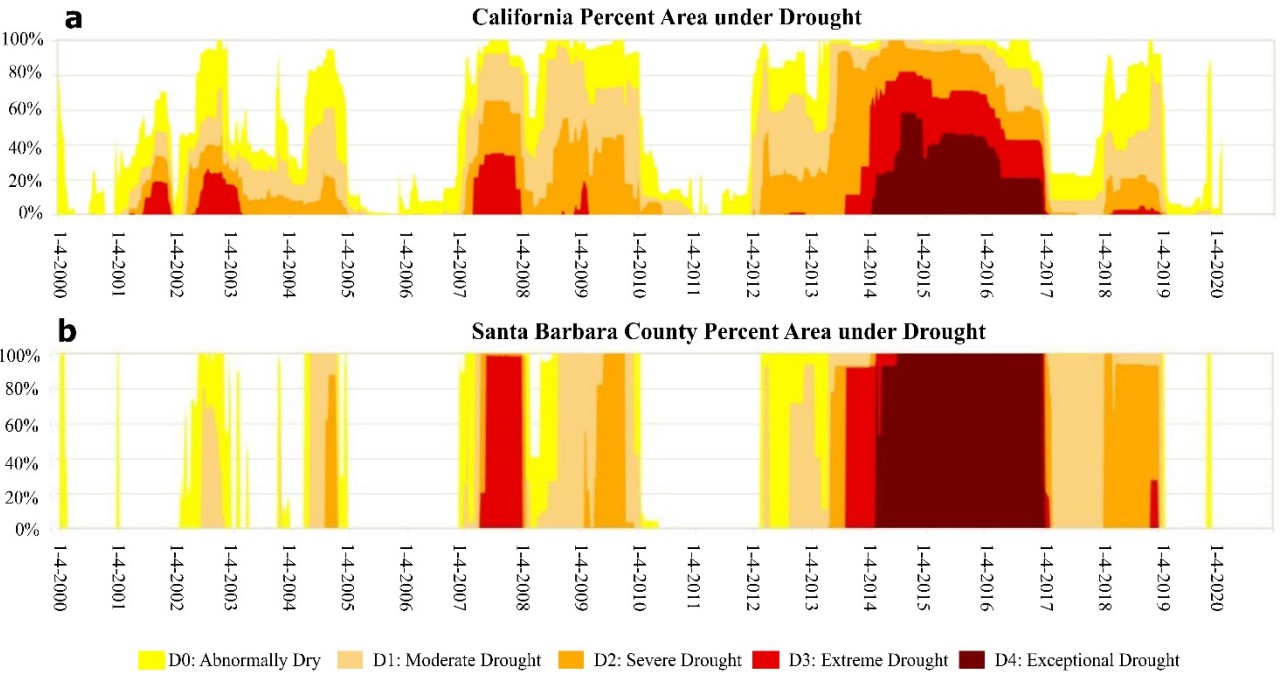

**Figure 2**

[caption] Figure 2: Timeseries of (a) percentage area of California under drought and (b) percentage area of Santa Barbara
County under drought. Santa Barbara was under a state of severe to exceptional drought between 2013 to 2019. [The U.S. Drought Monitor
is jointly produced by the National Drought Mitigation Center at the University of Nebraska-Lincoln, the United States Department of
Agriculture, and the National Oceanic and Atmospheric Administration. Map courtesy of NDMC.]





for several years (**Figure 2b**). By spring of 2017, the county was still under 'moderate' drought (D1), following a single wet winter season. However, the accumulated moisture deficit was high after several years of exceptional drought conditions, such that the state reverted to a state of 'severe' drought (D2) in 2018 after another abnormally dry year. The region finally came

out of the drought completely in early 2019 after the wettest rainy season since 2005. Based on these drought designations from the USDM, we defined the following periods for our drought analysis in SB County: Non-Drought (*PD*): 01-01-2008 to 31-12-2011 and Drought (*D*): 01-01-2012 to 01-01-2019.

## 2.4 NDVI Data

Vegetation indices from remote sensing have been widely used to monitor the effects of drought on vegetation, as well as the

links between precipitation, soil moisture, and plant sensitivity (Dong et al., 2019; Gu et al., 2008; Small et al., 2018). Multispectral indices, such as Normalized Difference Vegetation Index (NDVI), provide good spatial and temporal representation of drought conditions, which can be combined with in situ measurement of soil moisture for a more detailed understanding of drought propagation and drought stress on vegetation (Gu et al., 2008; Okin et al., 2018). To analyze the seasonality and relationship between soil moisture and vegetation we used NDVI derived from Landsat (Landsat-5 Thematic

Mapper, Landsat-7 Enhanced Thematic Mapper and Landsat-8 Operational Land Imager) to analyze the responses of grasslands to the multi-year drought in Southern California. Landsat provides images every 16 days with a 30-m pixel resolution. We defined polygons around the measuring stations to represent a broader area of homogenous grassland vegetation and soil textural properties at the coastal (19,800 m$^2$) and inland site (35,100 m$^2$). We quantified spatially-averaged NDVI over each polygon to create a monthly time series using median values of all pixels for each location for the period January 2008

to October 2019. NDVI, as a function of the visible and near-infrared wavelengths, ranges from +1 to -1 and reaches its maximum (saturated) value of 1 in conditions of high plant vigor and photosynthetic activity. Low or negative values are more representative of bare ground, senescent vegetation or water surfaces (Gillespie et al., 2018). Through a pixel-wise visual analysis of NDVI and comparison of different cover types (grassland, bare ground, forest, water) over our grassland sites, we established that green grassland vegetation is generally represented by values >0.3, while NDVI values <0.3 are more

indicative of brown or senescent (non-photosynthesizing) vegetation.





## 2.5 Soil Moisture Balance Model

### 2.5.1 Model Description

To characterize the effects of the recent California drought on soil moisture, we developed a simple, parsimonious model that
enabled us to better understand the linkages between climate, plant water availability and plant health and allowed for experimental manipulations of climate variables to explore plausible future climate scenarios. Rather than attempting to model detailed soil moisture processes, we used a soil moisture balance model (SMBM), which is based on a simple 'bucket' approach established by the FAO (Allen et al., 1998), and is a variant of a code previously developed for estimating groundwater recharge (Cuthbert et al., 2013; 2019). Simple modeling frameworks capable of linking vegetation to water availability can be useful
tools to assess past and future ecohydrological dynamics in a range of water-limited environments (Caylor et al., 2009; D'Odorico et al., 2007; Evans et al., 2018; Quichimbo et al., 2020). Model inputs include information on soil properties, vegetation cover and climate (precipitation and the meteorological variables required to estimate evapotranspiration (PET)) (**Figure 3**). Due to the flat topography of our study sites we assume runoff is zero, thus precipitation is either infiltrating into the soil or returned to the atmosphere through evapotranspiration.

The SMBM calculates a soil moisture deficit for a homogenous soil column of a certain rooting depth, which can be converted to vertically averaged, volumetric soil moisture content ($m^3/m^3$). The initial amount of available water to plants is defined by the water content at plant wilting point ($\theta_{wp}$ [$m^3/m^3$]), the effective field capacity ($\theta_{fc}$ [$m^3/m^3$]), and a certain rooting depth ($Z_r$ [mm]). Rooting depth is based on the dominant crop and was taken from Allen et al. (1998, Table 22) based on the dominant vegetation at the study sites. The corresponding relevant quantities of water available to plants are characterized as Total
Available Water and Readily Available Water. Total available water represents the amount of water a crop can extract from the root zone, depending on soil textural properties and rooting depth, while the readily available water represents a depleted fraction of the total available water that can be extracted from the root zone without the plants suffering from water stress. If the moisture deficit in the soil exceeds readily available amount, evaporation is adjusted through a water stress coefficient ($k_s$). As the water content gradually decreases through evapotranspiration following a rain event, the soil moisture deficit will
increase. If no additional moisture is added through more precipitation, soil water content will reach its minimum value at wilting point, where no water is left for evapotranspiration and $k_s$ becomes zero. The moisture deficit has reached its maximum value as the total available amount of water is exhausted. Following a heavy precipitation event, downward drainage (percolation) of water from the topsoil is occurring. No drainage occurs if the soil water content in the evaporation layer is below field capacity. A schematic overview of the key model parameters and the equations can be found in the Supplementary
Material (**Figure S1**).





For this study, we had information on soil properties available (**Table S 1**), however if field measurements are unavailable typical ranges for field capacity, wilting point, and rooting depths can also be found in the FAO56 Manual (Tables 19 and 22 in Allen et al., 1998). The depletion fraction factor (*pc*) that decreases the total available amount of water is generally dependent on vegetation/crop type and was set to a commonly used range between 0.2-0.6 (Allen et al., 1998, Table 22). The SMBM 240 was driven by precipitation from meteorological data and potential evapotranspiration (PET [mm/d]) was estimated through

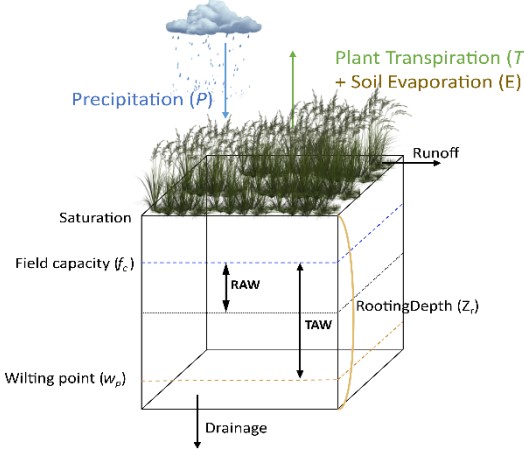

**Figure 3**

[caption] Figure 3: The Soil Moisture Balance Model simulates vertically averaged volumetric soil moisture for a homogenous column of soil over a certain rooting depth (Zr). Plant available water is limited by field capacity and wilting point and the relevant quantities defined through Total Available Water (TAW) and Readily Available Water (RAW). Runoff is assumed to be zero. due to the flat surface elevation at our sites.

| Parameter | Definition | COPR | AIRS |
|---|---|---|---|
| $Z_r$ [mm] | Rooting Depth | 700 -1000mm (both) | |
| $k_c$ [-] | Crop coefficient | From NDVI (both) | |
| $\theta_{FC}$ [m3/m3] | Field capacity | 0.4-0.5 | 0.2-0.3 |
| $\theta_{WP}$ [m3/m3] | Wilting point | 0.1-0.2 | 0.001-0.05 |
| Pc [-] | Depletion fraction for RAW | 0.2-0.6 (both) | |

[caption] Table 1: Ranges of used key parameter values used as input for the SMBM. Rooting depth is an estimate, based on dominant vegetation type at the sites. Ranges for soil textural properties are from Table S1 in the supplementary material. Crop coefficients are derived from NDVI for both sites. The depletion fraction is based on a standard range used by the FAO (Allen et al., 1998).



Penman-Monteith, using meteorological data from the weather stations. Other key parameters in the SMBM are shown in **Table 1**.

The FAO approach provides different methods for quantifying actual evapotranspiration (AET) for agricultural crops. In this study we used a time varying (dynamic) crop coefficient ($k_c$), which allows for variations in soil evaporation and plant
transpiration. It is therefore meant to represent the dynamics of vegetation and ground cover during the growing season, from initial emergence to full plant maturation to senescence. As vegetation cover evolves, the proportion of transpiration increases, until most of ET comes from transpiration and bare soil evaporation becomes negligible (Allen et al., 1998). Similarly, when vegetation is starting to senesce and photosynthetic activity is reduced, transpiration becomes negligible and evaporation from the (soil) surface increases. This dynamic approach allows the model a certain flexibility for a wide range of vegetation cover
types and gives a more representative estimate of evaporative losses. However, it requires an estimate of $k_c$ for the calculation of ET at each time step, which we obtain through remote sensing data.

### 2.5.3 Dynamic Vegetation Response

Previous studies have explored the relationship between multispectral indices, such as NDVI and crop coefficients, and have applied it successfully to estimate $k_c$ at the field scale for different locations and climate conditions (Hunsaker et al., 2005;
Kamble et al., 2013). The approach is based on the relationship between $k_c$ and NDVI, whereby the increase of $k_c$ from its minimum value at senescence to its maximum value at full plant growth is regressed against minimum and maximum values of NDVI for local vegetation (Kamble et al., 2013). For our study sites, oat grass (*Avena fatua*) is the predominant species, which is similar in growth and phenology to common oat (*Avena sativa*). Therefore, we used the FAO values of $k_c$ of common oat (Allen et al, 1998, Table 22) as ground-truth data to establish a relationship between minimum and maximum $k_c$ (0.3 and
1.15) and the median minimum and maximum NDVI of both sites (0.2 and 0.74). We developed our own relationship through linear interpolation of min/max kc and NDVI between monthly NDVI and $k_c$ for our two study sites:

$$k_{c, NDVI, i} = 1.57 * NDVI_i - 0.01 \qquad (1)$$

where NDVI is a monthly median value and $k_c$ is a temporally variable monthly value that represents the varying vegetation responses to climate as indicated by the NDVI signal. This equation is valid for NDVI in the range of 0.2 to 0.74, which
corresponds to the median minimum and maximum NDVI over the 11-year study period for both sites.



### 2.5.4 Model Implementation

Data for the period from January 01, 2008 to September 30, 2019 were used to calibrate the model. The range of estimates of SMBM parameters were constrained based on field measurements of soil texture (**Table S1**) and general estimates (rooting depth, Table 22 in Allen et al., 1998). We developed an envelope of uncertainty based on Monte Carlo sampling (1000 simulations from a uniform distribution) from these ranges. The data were separated into calibration and validation sets and model performance in each period was evaluated for acceptance or rejection of models. During calibration, model performance was optimized using data from January 01, 2008 to December 31, 2014. This time frame was chosen to include the natural variation of soil moisture dynamics, including non-drought and drought period. The model was then tested against data from January 01, 2015 to September 30, 2019. This period also includes natural variations in soil moisture, including the drought, individual very wet and dry years. Thus, we account for the possibility of different combinations of parameter values that may all be equally successful at reproducing the observed soil moisture. data We defined the quantitative measures of acceptance/rejection criteria using Kolmogorov-Smirnov (goodness of fit) testing to identify parameter combinations that achieve statistically similar (p > 0.01) distributions in observed versus simulated soil moisture. The temporal dynamics of soil moisture were evaluated via Nash Sutcliffe Efficiency (NSE) to identify parameter combinations that adequately simulated the observed soil moisture series (NSE > 0.5). The models accepted during calibration and validation periods were then evaluated via goodness-of-fit and the best model and its parameters was used for simulating soil moisture under simple climate change scenarios. We also included ±1 standard deviation of all accepted models in the results to show the range of working models.

### 2.5.5 Simple Climate Change Scenarios

Projections of future climate change in California suggest that there will be shifts in precipitation frequency and variability during the dry season, partly offsetting any increases in winter precipitation and possibly shifting towards more extreme events with an increased number of dry days and increased evaporative demand (Aghakouchak et al., 2018; Berg & Hall, 2015; Cook et al., 2015; Pierce et al., 2018). The projected rise in temperature is corroborated throughout the Southwest and across the entire continent (Diffenbaugh et al., 2015). Furthermore, trends in emissions for California point towards a higher emissions scenario of RCP 8.5, where annual maximum temperatures are projected to increase by more than 4°C (Thorne et al., 2016). Such increasing temperature projections are anticipated to have important implications for evaporative demand and soil drying, especially in such arid grassland ecosystems of Southern California.

We used the SMBM model to explore the possible effects of such variations in P and PET on soil moisture and vegetation in a simple parsimonious way, based on projections of shifting precipitation variability and evaporative demand (Berg & Hall, 2015; Pierce et al., 2018). In these explorations of climate change, we used monthly input data and did not alter other key





parameters, such as soil properties and vegetation cover. The approach of only altering P or PET forcing of the SMBM allowed us to separately explore the influence of changes in precipitation and evaporative demand to moisture and plant water availability, as well as vegetation growth under scenarios of more extreme drought. We developed three simplistic climate change scenarios based on regional climate change projections. The period 2012-01-01 to 2018-12-31 was used as a reference climate, and the climate scenarios are represented as a deviation from it as follows:

300         Scenario A) Simulates the effects of a prolonged dry season and a truncated rainy season from November – February. This scenario represents an extreme decline in annual precipitation totals, the loss of precipitation in the shoulder seasons and prolonged dry periods.

        Scenario B) Simulates redistribution of historic precipitation totals over a truncated rainy season from November – February, thus increasing the precipitation intensity and frequency during the compressed rainy season, combined with an
increase in dry season length. Projections of CMIP5 indicated an increase in the number of dry days combined with increased frequencies of heavy precipitation, overall increasing interannual precipitation variability over California (Berg & Hall, 2015).

        Scenario C) Simulates the effects of increased evaporative demand. Annual evaporative demand was increased by 10%, while precipitation frequency and intensity remain unaltered from the historical baseline. This scenario is consistent with a 4°C projected change in temperature for Southern California and much of the Southwest (Cook et al., 2015) under RCP 8.5.

In addition, we retained dynamic vegetation responses in our investigation of the climate scenarios. To replace historic NDVI values (which do not exist for potential future scenarios), we developed a heuristic relationship between NDVI and net precipitation (netP = P – PET) and evaluated the time period over which netP most strongly influences vegetation responses (1,2 or 3 months), based on correlation strength (Pearson's correlation). We used a power law to establish a final regression form and selected a correlation model that best explained NDVI variation, as a result of netP, considering drought and non-
drought periods separately, and used coefficient of determination ($R^2$) and root-mean-square-error to evaluate the chosen regression The relationship was used in the climate change simulations to create a synthetic NDVI input based on artificial netP simulations, which was then used to estimate $k_c$ based on Equation 1 and drive the SMBM.





## 3. Results

### 3.1 The Meteorology of the Drought

The 2012-2019 drought in California was marked by several years of above average temperatures, high evaporative demand, and low precipitation in Santa Barbara County. The seasonal differences during the March – October season between non-drought and drought periods was +1°C at the coastal site and +1.15°C at the inland site, with daily maximum temperatures during the March – October season being 6.2°C warmer at the inland site than at the coast. Temperature differences were significantly higher during the drought at both sites (KS =0.15 at the coastal and 0.0098 at the inland site, p=5.09$^{e-5}$) (**Figure 4a**). Due to the moderating effects of cooler/moister oceanic air and coastal fog, relative humidity at the coastal site averaged at 77%, with no significant changes between non-drought and drought periods (**Figure 4b**). Inland, the humidity was significantly lower during the drought (KS = 0.058, p=0.003), averaging 49%. The more moderate temperatures and high relative humidity at the coastal site were also reflected in a lower evaporative demand, resulting in ~50% lower PET compared to the inland site (**Figure 4c**). Monthly PET averages at the coastal site were 87 mm/ month and 152 mm/month at the inland site over the entire study period, with significant increases during the drought (KS = 0.091, p=5.11$^{e-5}$ coastal, KS= 0.093, p=

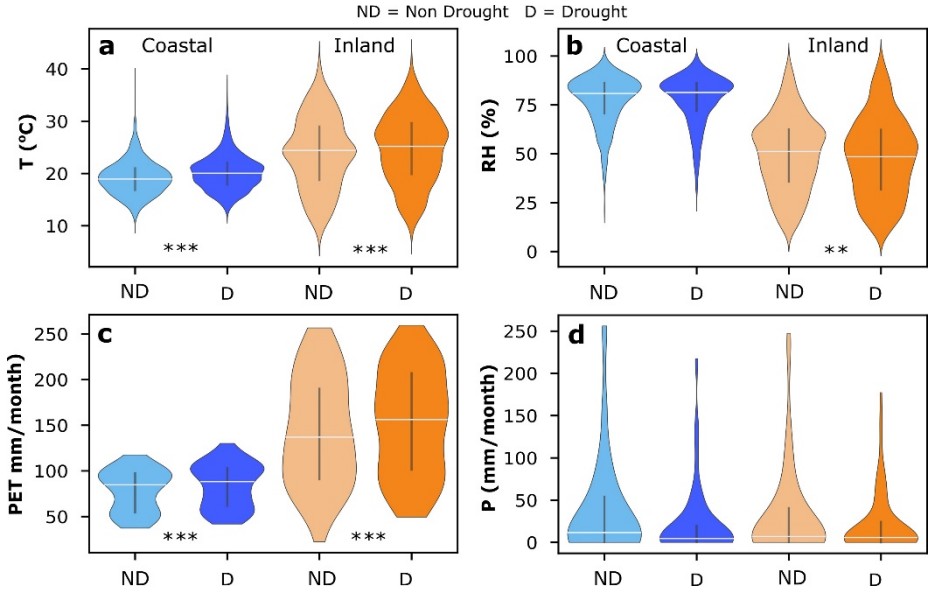

**Figure 4**

[caption] Figure 4: Violin plots of the climate variables a) temperature (T), b) relative humidity (RH), c) calculated potential evapotranspiration (PET) and d) monthly precipitation (P) during non-drought (light colored) and drought periods (dark colored) at our study sites. The vertical black line indicates the interquartile range. Medians are indicated by the white horizontal line. Statistical differences are indicated at the 0.01 (**) and 0.001 (***) level.





5.10$^{e-5}$ inland). Historical annual precipitation over the 11-year period was on average 20% less at the inland site than at the coast, as the site lies in the rain shadow of the Santa Ynez mountain range. The lowest October-September totals at both sites were recorded in 2014 with 170 mm/year at the coastal and 162 mm/year at the inland site (**Figure 4d**). however there was no

statistical significance in annual rainfall totals between non-drought and drought. High winter rains (382 mm/yr inland and 488 mm/yr coastal) occurred in 2017, but the area remained in a state of severe drought until early 2019. A single dry year in 2018 temporarily increased the drought stress on the region again, before a very wet rainy season in 2019 finally relieved the pressure on ecosystems and water resources in Santa Barbara County locations and the entire state (**Figure 2b**).

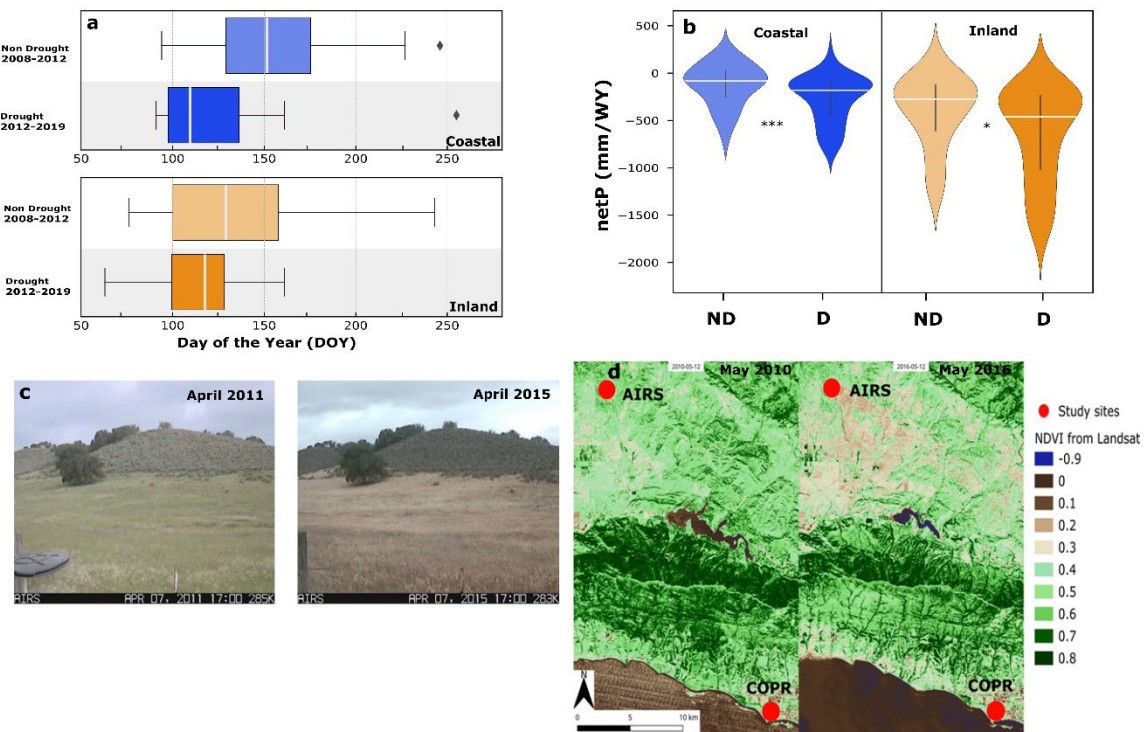

**Figure 5**

[caption] Figure 5: a) The onset of the dry season, presented as day of the year (DOY). The vertical white lines represent the median DOY of dry season onset and showed a shift towards an earlier onset during the drought. The whiskers indicate the maximum and minimum recorded DOY of dry season onset. Diamond marks represent outliers. b) Violin plots showing netP (P-PET) in mm/WY (October-September). The black vertical line indicates the interquartile range and medians are shown as white horizontal lines. During the drought netP becomes negative, indicating an increasing moisture deficit, due to increased evapotranspiration and reduced precipitation. Statistical significance of netP between drought and non-drought periods is indicated at the 0.001 (***) and 0.05 (*) level. c) The early onset and decline in greenness was visible at the inland site through webcam photographs in April 2011 and April 2015, and d) over the whole Santa Barbara region, as seen from NDVI images from Landsat in May 2010 and 2016.


While there was no significant difference in the amount of total precipitation, there was a shift in the onset of the dry season,
or the (spring) day of the calendar year after which no more precipitation was recorded for three consecutive months or more
until the start of the rainy season. During the drought, the dry season began at least one month earlier than during the non-
drought period (**Figure 5a**). At the inland site, the onset of the dry season shifted from a median DOY 130 to 117, which
translates to a temporal shift roughly from mid-May to mid to late April. At the coastal site, the median DOY shifted from 151
to 109 during the drought, i.e. from late May to early April. This shift triggered visible vegetation browning during the drought
by early April at the inland site, as opposed to a more gradual browning between May and June in the years preceding the
drought (**Figure 5c,5d**). Furthermore, increased evaporative demand and reduced precipitation during the drought also resulted
in a significantly lower netP (KS = 0.351, p = 0.00075 coastal, KS=0.253, p = 0.033 inland), implying limited water availability
for infiltration and soil moisture, especially further inland.

### 3.2 Soil Moisture and Plant Responses to Drought

The drought was expressed differently in the soil moisture at each site. Soil moisture observations showed increased drying of
soils during the drought at both sites, compared to the non-drought period, reaching extremely low moisture levels in 2013 and
2014 (degree of saturation fell below 5% inland) (**Figure 6a**). Similar low soil moisture occurred at both sites in 2008, a
particularly dry year for the SB region (**Figure 2b**). At both sites, the median degree of saturation was significantly different
between the non-drought and drought period (KS=0.12, p=5.09$^{e-5}$ coastal, KS=0.224, p=5.09$^{e-5}$ inland).





Monthly NDVI values reflect the strong seasonality of annual grass cover in the region, with a marked green-up period after

the winter rains, followed by a decline into brown conditions over the dry season. Although the median NDVI values were not

significantly different between non-drought and drought periods at either site (KS=0.14, p=0.54 coastal, KS=12, p=0.65

inland), there was an an increased variability between green up and vegetation die-off during drought. In particular, there was

a rapid increase of greenness during the drought post winter rains in 2015 and 2016, followed by an unusually rapid and early

decline of greenness in spring (**Figures 6c;6d**). Surprisingly, NDVI reached maximum values in 2015 at the height of the

drought (0.70 and 0.77 coastal and inland, respectively). It is notable that the drought NDVI peak values are higher than those

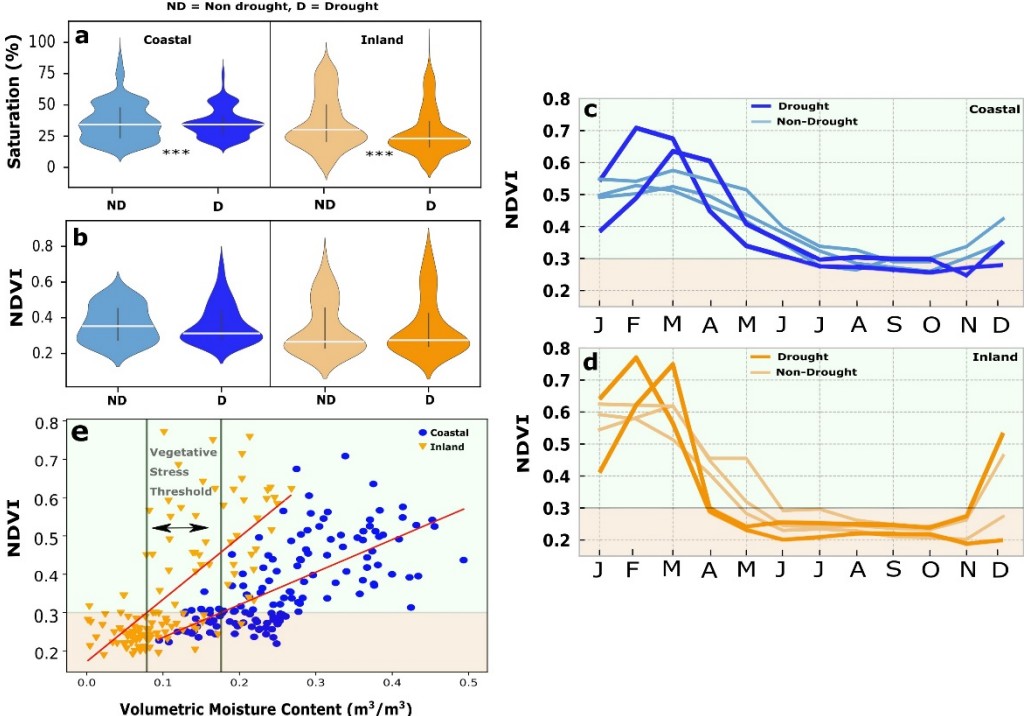

**Figure 6**

[caption] Figure 6: a) Violin plots showing the degree of saturation during non-drought and drought periods at the coastal (blue) and inland site (orange). Vertical lines indicate the interquartile range and medians are indicated by a white horizontal line. Statistical significance between the two periods is indicated at the p<0.001 level (***). b) Violin plots of monthly median NDVI distributions accentuate the differences between the two sites. c) and d) Monthly median NDVI during drought years 2015 and 2016 (dark solid lines) and non-drought years 2010, 2011, and 2019 (light solid lines) for the coastal (blue) and inland (orange) site respectively. e) The relationship between NDVI and measured volumetric soil moisture was used to establish a vegetative stress threshold at each site that indicates moisture conditions more conducive to senescent vegetation.





for the non-drought period at both sites, but they were very short-lived. NDVI dropped steeply back to low values, in contrast to the shoulder of greenness and slower decline of NDVI that occurred in most non-drought years (**Figure 6c;6d**).

During the drought, NDVI dropped rapidly below 0.3 in April at the inland site, supporting the result seen in the visible

browning through webcam images and NDVI over the region (**Figures 5c;6c;6d**). These differences in the seasonal variation of NDVI suggest an aggressive strategy of grass green up after winter rains, accelerated by mild winter temperatures during the drought, especially during the exceptionally warm winter in 2014-2015. The growth of additional vegetation under these conditions likely led the observed rapid decline in moisture during spring, as vegetation quickly depleted any excess moisture, and subsequently the increased browning and senescence due to the early onset of the dry season. Pearson correlation between

NDVI and soil moisture of the concurrent month was strongly positive and statistically significant for both sites ($R^2 = 0.68$ coastal and inland, $p < 0.001$). The correlation between the two variables was used to establish an heuristic vegetation stress threshold at VMC = 0.17 $m^3/m^3$ for the coastal and VMC= 0.07 $m^3/m^3$ for the inland site, which we associated with very low rates of photosynthetic activity, based on an NDVI threshold of 0.3 (**Figure 6e**).

Pearson correlation between NDVI and netP revealed a three-month lag in netP and NDVI at the coastal site ($R^2=0.82$), while

at the inland site it was a two-month lag ($R^2=0.74$). In order to develop a predictor (leading indicator) of vegetation response to netP, we fitted regression models as follows:

$$NDVI_i = a * b^{netP_m} \qquad (2)$$

where $NDVI_i$ denotes predicted monthly NDVI, $netP_m$ is an amount of netP accumulated over a number of months m, and a and b are regression coefficients. The results of regression coefficients and correlation are shown in **Figure 7** for each site

during the drought and non-drought periods.

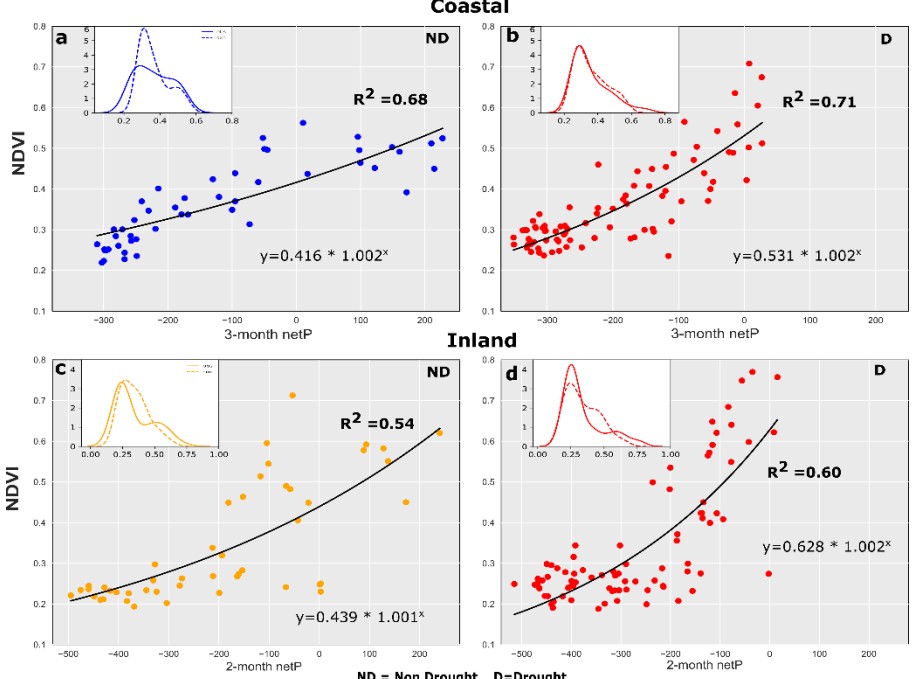

**Figure 7**

**[caption] Figure 7: Power law relations between a) 3-month netP and NDVI for the coastal site during the non-drought and b) drought period; and between c) 2-month netP and NDVI at the inland site during the non-drought and d) drought period. Distributions between observed and simulated NDVI for the historical period are shown in the insets. $R^2$ denotes the strength of the regression model.**

### 3.3. Soil Moisture Water Balance Model Performance

Given the simple structure of the SMBM, we were encouraged that the best models at each site were effective at capturing and predicting the timing and magnitude of interactions between P, PET, and soil moisture (**Figure 8a-b**). Kernel density estimates (KDE) for observed and simulated soil moisture distributions were also statistically similar (**Figure 8c**; KS=0.086 and p=0.68 coastal and KS=0.14 and p=0.18 inland at p> 0.01). However, we note that best-fit, simulated soil moisture at both sites may over- or under-estimate observed VMC at particular points in time. Notably, the best model from the Monte Carlo simulations at the inland site was not able to capture the extreme dryness in 2013 and 2014. The SMBM assumes plant wilting point is the lowest level of soil moisture. However, in reality soil moisture may decline below wilting point during the extremely dry



periods. In such conditions, senescent or even dead plants can still act as a medium for the transference of water long after wilting has occurred, potentially compounding the effects of soil drying by evaporation (Briggs and Shantz, 1912).

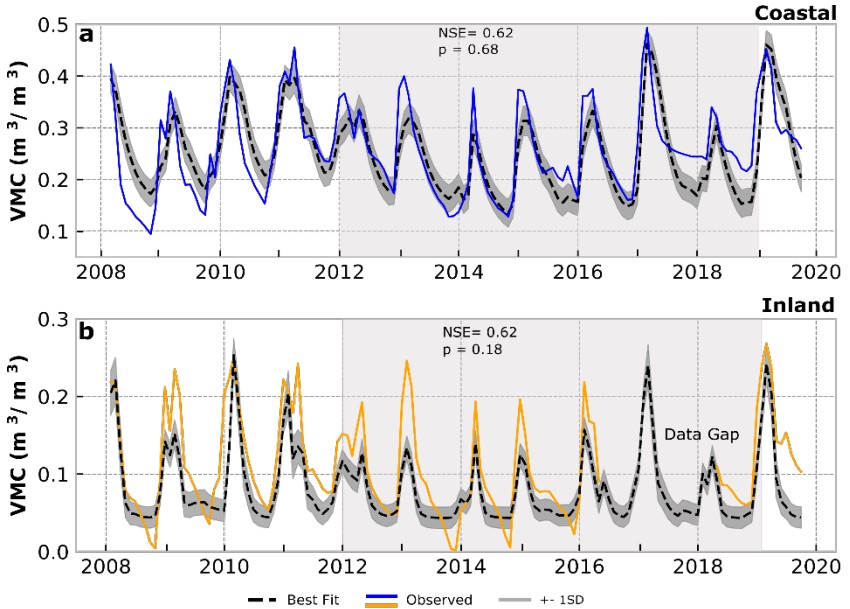

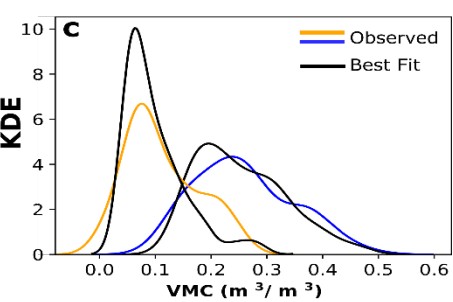

**Figure 8**

**[caption] Figure 8: SMBM results for the a) coastal and b) inland site. Observed soil moisture is indicated in a solid line (blue-coastal, orange-inland), while simulated moisture is shown with a dashed black line. Grey shaded banding indicates ±1 Standard Deviation (SD) base on the output of 1000 Monte Carlo simulations. c) KDE curves of the observed and simulated moisture for the best model fit.**

### 3.3 The response of soil moisture to plausible climate change scenarios

Over the historic time frame studied here, modelled soil moisture at the coastal site was typically below the vegetative stress
threshold, resulting in very low rates of photosynthetic activity (**Figure 6d**), prior to the onset of winter rains. Scenario A (truncated rainy season with precipitation occurring only between November to February) represents the loss of spring and early summer precipitation events, which results in a prolonged period of vegetative stress due to low soil moisture, beginning earlier in May. (**Figure 9b, coastal**). Scenario B (redistribution of the historical annual precipitation over a truncated rainy season between November to February), increases the intensity of rainfall events and thereby extends the duration of peak
moisture levels in months of winter rainfall (broadened peaks in 2013 and 2015 and high associated VMC between January –





April, **Figure 9c, coastal**). This result indicates lasting benefit from the simulated increased precipitation intensity, followed by several dry months of vegetative stress. Scenario C (increase of annual PET by 10%, reflecting ~4°C increase in annual temperature) drives a net reduction of soil moisture throughout the entire period and also an increase in the duration of the period with moisture levels below the vegetative stress threshold (**Figure 9d, coastal**). The increased climatic water deficit in

**Figure 9**

[caption] Figure 9: Simulated soil moisture at the coastal (left) and inland (right) site under a) the historical drought with grey shading indicating ±1SD from 1000 Monte Carlo simulations, b) a simulated truncated rainy season from November – February, c) a simulated redistribution of precipitation totals within a truncated season and d) a simulated 10% increase of annual PET. Green shading indicates moisture more conducive to photosynthetic activity, while brown shading indicates low soil moisture indicative of browned and stressed vegetation. The gray horizontal line indicates a vegetation stress threshold established for each site. Red bars indicate P loss, green bars indicate P gains compared to historic conditions. e) Violin plots of simulated soil moisture with medians indicated by a white horizontal line. The black vertical line indicates the interquartile range. Statistical significance from historical simulations is indicated at the p<0.001 level (***)



this scenario could be conducive to a shifting baseline soil moisture towards drier conditions on average and drive coastal areas into more extreme and prolonged drought periods. Each scenario for the coastal site resulted in a strong statistical difference with the historic simulated moisture distribution (Scenario A: KS = 0.73, Scenario B: KS=0.63, Scenario C: KS=0.65 at $p<0.001$), underlining the soil moisture effects of extreme drought scenarios.

Because of the warmer and more arid climate further inland, simulated changes in precipitation frequency and intensity further

exacerbated the intense soil drying observed during the historical drought period (**Figure 9a, inland**). Scenario A created frequent occurrences where soil moisture would only reach peak levels briefly during and directly after the winter rains of particularly wet winter seasons (2013, 2015, 2016, 2017, **Figure 9b, inland**). In drier winters (2014, 2018), soil moisture only received minimal resupply and thus remained below the vegetative stress threshold. In Scenario B, the soil benefited from the simulated increased precipitation intensity during the winter. However, the early onset of the dry season and lack of spring

precipitation pulses still resulted in simulated moisture levels dropping below the vegetative stress threshold for several months, creating an extended dry season **Figure 9c, inland**). Scenario C showed the effects of simulated increased evaporative demand on an already semi-arid landscape (**Figure 9d, inland**). The increase in evaporative demand would be sufficient to reduce simulated soil moisture to an increasingly dry stage from April onwards, with no moisture recharge despite precipitation later in the year because evaporative demand exceeds precipitation inputs. The simulated moisture scenarios for the inland site

are statistically different from to the historical simulations (Scenario A: KS= 0.65, Scenario B: KS=0.60, Scenario C: KS=0.60 at $p<0.001$).





## 4 Discussion

In light of the progression of climate change in semiarid environments, it is essential to understand how lowland grassland
ecosystems responded to the recent multi-year drought in Southern California. This would enable predictions and/or
anticipation of how soil moisture and grassland dynamics might respond to intensified moisture limitations under future
scenarios of climate change across the region. Understanding the climatic drivers of shifts in soil moisture and water
availability to vegetation (and correspondingly, to the health and functioning of grassland ecosystems), are fundamental for
managing and ameliorating the negative impacts of climate change. The severity of the recent synoptic California drought and
its effects on vegetation were most notably documented through upland forest canopy water stress and mortality (Asner et al.,
2016; Fettig et al., 2019; Goulden & Bales, 2019), as well as through declining groundwater levels that heavily impacted
agricultural production throughout the Central Valley (Thomas et al., 2017; Xiao et al., 2017). Similarly, the intensified
moisture loss and accelerated ET also impacted lowland vegetation in Southern California, including differential species
responses within chaparral and grassland ecosystems (Breshears et al., 2005; Gremer et al., 2015; Okin et al., 2018; Wilson et
al., 2018). While the landscape in Southern California is dominated by vast stretches of brown grasslands during the dry season,
the 2012-2019 drought in Santa Barbara Country was so intense and prolonged compared to the rest of the state (**Figure 2**),
that it propagated into multiple years of soil moisture deficits, early die-off of grasses **(Figures 4-6)**, and an overall drier
landscape primed for fire. Previous studies have noted that increased greenness during spring exacerbated soil drying in the
summer (Lian et al., 2020) and our study showed that the above average temperatures in combination with changes in the
deliverance of winter precipitation led to an early and increased vegetation green up which resulted in accelerated soil drying
in spring and unusually early senescent vegetation by April.

Our analysis revealed that winter/spring precipitation deficits, coupled with higher evaporative demand in Southern California
led to temporal shifts in the onset of the dry season, which led to increased soil drying in spring and summer. The loss of
essential precipitation pulses in spring months generated soil moisture deficits and faster die-off (browning) of grasses. Our
modeling analysis further highlighted connections and feedbacks between climate and grassland vegetation in Southern
California, which may be sensitive to climate change. Our findings suggest that such vegetation communities, widespread over
the region and more broadly over the Southwest and other Mediterranean climate systems, are becoming increasingly
vulnerable to climate change that favors milder winter temperatures and increased precipitation variability, thus increasing
drought frequency and magnitude. In arid regions, the capacity of soils to store water becomes increasingly important as soil
moisture can mitigate the impact of drought on grassland vegetation health (Gremer et al., 2015). The results from contrasting
sites (coastal and inland) corroborate studies showing differential responses to the drought over short distances due to spatial
variation in soil texture (Gremer et al., 2015; Liu et al., 2012; Okin et al., 2018) (**Figures 5;6**). Such differences in soils between





our sites, despite similar vegetation, produced different responses to the driving climate. For example, the lower water holding capacity and higher evaporative demand of the inland site led to much earlier senescence, selectively priming it for large and destructive wildfires and likely bringing these ecosystems to their physiological limit. This result can be viewed alongside prior work in the Southwest suggesting that chapparal landscapes (Okin et al., 2018) and perennial ($C_4$) grasslands (Gremer et al., 2015) are increasingly prone to negative impacts from drought. With a broader perspective of responses to water limitations in vegetation communities across arid and semi-arid lands in the Southwest, our results underscore that climate change impacts may differ substantially over short distances, thus affecting large areas of annual grasses, perennial grasses, and chapparal vegetation in more arid environments beyond their ability to adapt to these shifts, potentially priming increasingly large areas for widespread wildfires and increase the risk of widespread desertification of the landscape.

Grassland habitats in arid and semiarid environments are especially sensitive to changes in evaporative demand, amount and timing of precipitation and soil moisture availability (Gremer et al., 2015; Liu et al., 2012; Munson, 2013; Peters and Yao, 2012). The 2012-2019 California drought exemplified many of the potential deleterious effects of climate on vegetation communities (a truncated rainy season, increased evaporative demand, and longer dry periods), which led to an early die-back of annual grasses. Given how widespread this last drought was in terms of spatial footprint and temporal length, the effects of such drought conditions could be devastating to perennial grasses and chapparal communities with larger consequences for entire grassland/shrubland ecosystems (Gremer et al., 2015; Okin et al., 2018; Petrie et al., 2015). If these sorts of drought occur in the future and if they intensify, it is expected to lead to dramatic shifts in soil moisture towards a drier baseline for extended periods. This outcome could potentially leave grassland species diversity and productivity severely compromised, due to increased susceptibility to encroachment of invasive species and expansion of shrublands, thus limiting native species regeneration (Bradford et al., 2020; Peters et al., 2010; Reynier et al., 2017). Earlier and longer dry seasons with increased temperatures and increased evaporative losses and soil drying also reduce surface evaporative cooling and increase the possibility of more intense summer heatwaves (Lian et al., 2020). We also might expect conditions more conducive to large and widespread wildfires due to greater and deeper drying of soils and vegetation, raising the impact of fire severity and extent to new unprecedented levels in an already fire-prone region (Westerling et al, 2006; 2007). The increasing intrusion of invasive annual grasses has been linked to an increase in lighter fuels, leading to conditions under which fires are cured earlier in the growing season (Abatzoglou and Kolden, 2011). Thus, more frequent and widespread disturbances through wildfire would increasingly foster conditions under which native grassland vegetation health and diversity may be detrimentally affected, thus furthering a fire-invasive species feedback loop and limiting native species regeneration (Abatzoglou and Kolden, 2011; Donovan et al., 2020; Reynier et al., 2017)



With climate change projected to impact the temperate and precipitation regimes in California and much of the Southwestern U.S., the frequency and magnitude of droughts and drought-like conditions is expected to increase (Bradford et al., 2020; Diffenbaugh et al., 2015). Under a more severe emission scenario of RCP 8.5, the frequency of extreme dry years is projected to almost triple and temperatures are projected to increase by up to 4°C throughout California (Pierce et al., 2018; Thorne et al., 2016). Precipitation projections remain uncertain (Pierce et al., 2018, Bradford et al., 2019), but given the degree of already existing aridity in the Southwest, even relatively modest changes to precipitation intensity and frequency would create conditions much more conducive to prolonged drought periods. The increased occurrence of extended dry periods would slow down the recovery from moisture deficit inherited from prior drought conditions, raising the potential for soil moisture deficits to accumulate to unprecedented levels (Diffenbaugh et al., 2015). Such changes to the climate system, partially represented by our modeled moisture scenarios through truncated rainy seasons (Scenario A), intensification of rainfall within a truncated rainy season (Scenario B), and an increase in evaporative demand under a 4°C change (Scenario C), suggest that soil moisture would remain below a level that supports vegetation growth (vegetative stress threshold) for extended periods of time, with only minimal moisture input during the cool season. Our results indicate that such changes to evaporative demand and precipitation would potentially exacerbate the negative effects of drought on coastal grasslands, as the increased loss of moisture may dampen the buffering effects of fog. The overall increase in aridity through increased evaporation losses across Southern California and the Southwest would put a new strain on these sensitive areas and leave them potentially unsuitable as climate refugia and habitats for critical threatened and endangered species in the future. Similarly, the increased loss of moisture from elevated temperatures in semi-arid regions further inland could induce widespread desertification of the landscape, an increasingly altered species composition and abundance and chronic moisture deficits, thus negatively impacting grassland ecosystems over a broad spatial extent.





## 5 Conclusion

505 The 2012-2019 drought in California had profound impacts on soil moisture and vegetation. Through long-term monitoring data we delineated the differential responses of soil moisture and vegetation dynamics of grassland ecosystems to the multi-year drought in Southern California. A temporal shift of dry season onset led to early senescence and browning of vegetation and rendered soil moisture resources prematurely exhausted and the landscape primed for easily ignited wildfires. During the drought, there were changes in the temporal patterns of vegetation productivity, including increased greenness attributed to mild winter temperatures after prolonged dry periods. However, this new vegetation growth quickly dried out due to the early 510 onset of the dry season, exacerbating the soil moisture deficit.

Through a simple, parsimonious soil moisture water balance model we further explored the dynamics and water balance in terms of soil moisture for grasslands under different conditions that represent possible future climate change scenarios. We linked soil moisture and vegetation response through NDVI and explored the effects of various climate change scenarios. The results suggest that changes to precipitation and evaporative demand could have unprecedented effects on soil moisture and 515 water availability to grassland ecosystems, leading to rapid die-back and prolonged desiccation of the landscape. In future, more extreme and prolonged droughts, characterized by a shorter rainy season, higher evaporative demand, and/or protracted dry periods, will likely lead to an increased soil moisture deficits, as moisture levels are likely to drop to a level of elevated vegetative stress for much of the year. The loss of precipitation pulses in spring and summer, a continuing shift of early dry season onset and increased evaporative demand are likely contributors to affect grassland ecosystems in future and drive even 520 previously buffered coastal areas into more severe droughts as well as induce widespread desertification of the landscape in more semi-arid environments. A shift to a drier moisture baseline of soils and vegetation could potentially have deleterious effects on species diversity, increase the risk of shrub encroachment and invasive species and leave the region overall more prone to destructive and widespread wildfires.



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



**Author Contributions:**

710 Conceptualization: MBS,MOC,KC,DR,JS

Methodology: MMW,MBS,MOC

Data Curation: MMW,RS

Formal analysis: MMW,RS

Writing – Original Draft: MMW

715 Writing – Review & Editing: MMW,MBS,MOC,KC,DR,JS,RS

**Acknowledgments and Data**

This work was supported by The National Science Foundation (BCS-1660490, EAR-1700517 and EAR-1700555) and the Department of Defense's Strategic Environmental Research and Development Program (RC18-1006). We thank D. Roberts for providing the IDEAS data set, which is publicly available at http://www.geog.ucsb.edu/ideas/. MOC gratefully
720 acknowledges funding for an Independent Research Fellowship from the UK Natural Environment Research Council (NE/P017819/1).

The authors declare that they have no conflict of interest.