# Peer review of "Onset and propagation of drought into soil moisture and vegetation responses during the 2012-2019 drought in Southern California"

_Hydrology and Earth System Sciences, 2020_

## Referee Comment (RC1) · Anonymous Referee #1 · 29 Oct 2020

The manuscript by Warter et al. "Onset and propagation of drought into soil moisture and vegetation responses during the 2012-2019 drought in Southern California" presents a comparative analysis of climatology, soil moisture, and vegetation characteristics at two sites in southern California. This manuscript builds on earlier CA drought studies linking moisture deficit/depletion with remotely sensed vegetation characteristics with a focus on the grassland ecosystem. Measurements of precipitation, relative humidity, soil moisture, and estimated potential evapotranspiration (PET) were compared and contrasted between "drought" and "non-drought" years and two sites. Fur-

ther, the authors have combined ground-based and remotely sensed measurements to calibrate and validate a water balance model to explore soil moisture evolution under a more "intense" future droughts. Overall, the questions posed in the manuscript are interesting and data/analysis presented supports the thesis (see few issues warranting attention below). However, the writing and presentation throughout the manuscript comes out as a little ambiguous and often redundant. A careful editing will help highlight the key points. Some points for consideration:

1) Delineation of "drought" and "non-drought" period is based on USDM data seems a bit random. 01-01-2008 to 31-12-2011 defined as a "non-drought" period but it contains periods of "Extreme" and "Severe" droughts. Similarly, 01-01-2012 to 01-01-2019 "drought" period contains drought-free days along with periods of "Extreme" and "Severe" droughts. Since this classification is a basis of the analysis that follows, a more robust classification, perhaps based on drought categories, is needed. 2) Analyzing and comparing PET and P between drought and non-drought periods, defined based on NMDC data, seems like going in circles since NMDC drought categories are derived from the very dataset. 3) NDVI derived from Landsat-5, Landsat-7, and Landsat-7 are not comparable and must be homogenized and filtered from clouds and other types of data noise (Goulden and Bales, 2019). I was unable to figure out if homogenization and cloud correction was performed or not. Also, considering the short growing season, a median NDVI value may not be appropriate as it may end up representing the NDVI at the beginning or end of the month. See Roche et al. 2018 https://doi.org/10.1002/eco.1978 for centering technique. 4) NDVI exhibits saturation beyond some threshold precipitation or available water, it can be seen in Figure 7a. You don't expect the NDVI to continue to increase with increasing water availability. Some vegetation expansion is possible when ample water supply is available and other resources (energy, nutrient etc.) are not limited but eventually max out. Fitting an exponential model ignores this fact. 5) The definition of polygons with homogeneous vegetation and soil textural properties requires further explanation. Considering the fact that you have a mixture of vegetation at both sites, how did you define "homogeneous"? 6) The scenarios can be better described in the methods, I could not understand Scenario A and B until looking at figure 9. What is the meaning of the truncated rainy season and how annual P from the truncated months are redistributed? Also, these scenarios represent intense future drought as posted in the research question (iii) but the presentation of results and discussion comes out as typical climate change scenarios. 7) Figure 9 is interesting but can be conceptually predicted without running a model. Perhaps these results can be analyzed to better understand the onset and longevities of the drought. Something similar to 5a but for different scenarios.

Minor points: 1) Suggesting removing the unnecessary background information from the methods, i.e. do we need introductory sentences like these "Soil moisture is essential for plant growth and -health and accordingly, there are strong seasonal responses of vegetation to temperature and precipitation (Coates et al., 2015; Roberts et al., 2010)" to describe the study sites? 2) Precipitation values reported on top of the page 7 don't match the 20% difference reported on top of page 17 3) You mentioned inland site is not used for grazing, how about the coastal site? 4) Provide mean temperature for the two sites 5) Table S1, note the data formatting issue 6) Shortwave and longwave radiation measurements: are these net radiations? 7) L155: "For each site, we extracted daily maximum daytime temperatures, humidity and precipitation totals and calculated monthly averages to define the meteorology of the drought"- not clear. Which variables are daily maximum and which ones are totals? What do you mean by the monthly average of precipitation total? 8) PET calculation using the Penman-Monteith model need more information on how other inputs were derived i.e. conductance, ground heat flux etc. 9) Stevens hydro probe, provide manufacturer and model 10) L166: here you argue for using the degree of saturation but then end up comparing VMC in Figure 9. Relative saturation may have been more appropriate as it accounts for differences in residual WC between the two sites. 11) Fig S1 SMD can be equal to RAW as stated in the text 12) Showing Fc, Wp, RAW, and TAW in figure 3 is misleading. The picture depicts a soil profile and not a unit volume. In its current form, it looks like the Wp is always at the bottom of the root zone. 13) Equation 1, I don't quite understand what

minimum and maximum Kc & median minimum and maximum NDVI means. Are not you regressing the monthly Kc values against monthly NDVI values with the index I being the month 1 through 12? 14) P-PET is not really a net precipitation, it is closer to aridity P/PET 15) L320 2012-2019 drought is only relevant for southern California. Statewide, the drought ended in 2016. 16) Fig. 8: At what depth these soil moisture measurements were made? Is the simulated VMC are for the same depth or integrated over the entire root zone?

Thank you for the opportunity and I hope you find these comments helpful.

---

## Referee Comment (RC2) · Anonymous Referee #2 · 9 Feb 2021

This paper discusses the impacts of prolonged recent drought in California on the vegetation and soil moisture dynamics of natural grasslands in two contrasting climatic regions of Santa Barbara Co, CA. This controlled and localized experiment leads to some interesting findings re. grassland response to drought, with ramifications for wildfire and species diversity under varying climate change scenarios.

I think this is an interesting and generally well written manuscript; however, some clarifications and potential changes in definition of drought vs. non-drought years are required. In addition, a few suggestions to enhance readability are given below.

[Figure]

General comments:

Not sure the title makes complete sense as written. Taken in parts... Onset of drought into soil moisture responses? Consider reworking, with a focus on natural grasslands.

In general, I would recommend reducing use of "we" and shifting to a more passive tone throughout the manuscript. Shifts the emphasis from what the authors did and thought to a more objective viewpoint.

Parts are overly wordy and could use some critical editing to pare down to what is really essential to move the key story forward. The first paragraph of the introduction, for example, discusses forest response to drought in depth before getting to grasslands, which is the focus of the paper. Revisit the wording of the first sentence – could read something like "resulting in substantial impacts to water resources and ecosystems. These impacts varied regionally, depending on climate, elevation and biome... For example, upland forests...[1 or 2 sentences]. The impacts of drought on California's grasslands have been less well studied..."

Specific comments:

L158: Add citation for the Penman-Monteith equation.

L165: Explain the choice of the shallowest soil moisture observations as reference in this study. Why not the 20 or 50 cm measurements, which may be more representative of the root zone?

Fig. 2: Can the time markers in this figure be at 1-1-YYYY rather than 1-4-YYYY (is that April 1)? Seems a little cleaner...and less ambiguous re. date format. Inclusion of 2008 and 2009 in the non-drought period is problematic, since most of SB County was under moderate-severe drought in those years. Why not separate out truly non-drought years, defined as years with some prescribed fraction in drought-free conditions, even if they are not temporally contiguous?

Sec 2.4: Discuss noise reduction and normalization applied to the NDVI data obtained

from Landsats 5, 7 and 8 to generate multi-year timeseries. Were TOA band values used, or surface reflectance?

Sec 2.5.1: The description of the Soil Moisture Balance Model could be shortened significantly.

L215: "...to estimate potential evapotranspiration (PET)..."

Table 1 caption: Remove first of two "used" in first sentence.

L253: This sentence does not read well. Is there a missing comma after NDVI?

Sec 2.5.4: Seems to be an inconsistency in stated calibration range between line 267 (2008-2019) and L 272 (2008-2014). Please clarify.

L339: It is surprising that there is no difference in precipitation between drought and non-drought years. Perhaps a more stringent separation of these years would yield greater difference?

L366: What is meant by "aggressive strategy"? Maybe a use a different term.

L 451-452: The difference in response isn't only due to soil texture, right? Difference in climate (aridity) also drove response.
* * *

---

## Author Response (AR1)

We thank the Editor for giving us the opportunity to revise our manuscript to address reviewers' comments. We found the comments very helpful, and they led to a substantial improvement to our paper, including its overall presentation and impact. Below we provide point-by-point responses to comments by the Editor and both reviewers (in bold). We also describe the specific changes made to the manuscript compared to the originally submitted version. The line numbers below refer to the updated manuscript.

We believe the entire review process has greatly improved the manuscript and we once again thank both reviewers, and the editor for their time and effort.

**Editor Comments and Description of Major Changes to the Manuscript**

Dear Authors,

Your original submission was evaluated by two experts who rated it fairly well, albeit raising some concerns. From your responses to both reviewers during the discussion step, I see the potential for improvements, and I release the paper for revisions. As also raised by one reviewer, I suggest you should improve the way your study is presented. It is desirable to have more concise but effective introduction and methods sections, while focusing more on a much better presentation and discussion of the outcomes of your study.

We took on board these comments from the Editor and the reviewers, which suggested improving the presentation of the paper. We looked at this in detail and found many places to improve the manuscript in the introduction, results, methods, and discussion. We also added a figure to the paper that better contextualizes the results from our water balance modeling (new Fig 9). We provide details on these changes below with line number indicators. This section is followed by the point-by-point responses posted as author comments online. We also note that, during our revision, we identified an error in the submitted Fig 9, which has now been fixed in the updated version of that figure (current Fig 8). Finally, we realized that one of the submitted figures (Fig 7) was not necessary to include in the main paper, so we have moved it to supplemental material.

**L 30:** We added additional material to the abstract to highlight some key findings that resulted from rerunning our simulation using the updated approaches.

**L83:** Following the editors' suggestion, as well as the reviewers', we improved the introduction section by deleting unnecessary passages and moving an introductory sentence from the study site description to the introduction.

**L208:** This line was moved to the introduction for better fluency of the text.

**L302:** We added this sentence to clarify the purpose and approach of this study.

**L324:** Reviewer 1 suggested this sentence to be removed from the site description. Instead, we placed this sentence in the introduction section.

**L365:** As suggested by both reviewers, we reevaluated our drought designations and now present results classified into three categories: no drought, moderate drought, and extreme drought based on the USDM categories. This change has been addressed in detail within the comments to reviewer 1 (see below).

**L469:** We switched the order of the sections describing our data and the section on NDVI for more logical flow of information.

**L500:** Following both reviewers' comments, we homogenized the NDVI data according to the approach suggested by Reviewer 1, and we provide additional information on the process. This is also addressed below.

**Sect. 2.5** Following the suggestion of Reviewer 1, we updated and simplified Fig. 3 to a 2D representation of a soil column and the processes therein. We also shortened the model description as suggested by reviewer 1 (addressed below).

**Sect. 2.5.3** As per the reviewers' suggestion, we reevaluated our approach to quantify dynamic vegetation response. We appreciate the opportunity to do so, as this has improved our results. We followed the approach by Glenn et al. (2011) that uses NDVI as a proxy to estimate crop coefficients. This is addressed in detail below in the comments to reviewer 2.

**L714:** We adapted Scenario C to better represent the combined effects of extreme drought conditions consisting of a truncated rainy season, decreased rainfall, and increased evaporative demand. The scenario now includes a truncated season with reduced P intensity (-25%) as well as increased PET (+25%), representing a potential +4°C increase in annual temperature. We ran our climate simulations with this updated scenario, which can be seen in the new Figure 8.

**Sect. 3.1, 3.2:** Following the suggestions from both reviewers and the implementation of the new drought categories, all plots in this section have been updated to include new drought designations. The statistical differences and mean values have been adapted in the text accordingly. We also provide an additional table in the supplementary material with all the relevant statistical parameters and information.

**L887:** We slightly changed our approach here and decided to use a linear regression instead of a power law fit between available precipitation (aP) and NDVI. Also, to enable future simulations without a priori assumptions of drought conditions, we established one relationship between NDVI and aP for each of our sites as predictor of vegetation response to precipitation deficit under plausible future climate scenarios. As per Reviewer 2, we implement a maximum NDVI threshold to account for saturation of the NDVI signal (addressed in comments below).

**Fig.7**: We changed the presentation of this figure to saturation (%) as reviewer 2 suggested.

**Sect. 3.3:** We reworked this section and the overall presentation and discussion of our results. As part of this effort, we redid our simulations using the updated approaches to estimate aP, NDVI and kc (outlined below), and produced updated versions of the submitted Figures 8 and 9 (now Figs 7 & 8). We also produced an additional figure (new Figure 9) and associated discussion that plots cumulative water balance outputs from the model, which provides a stronger context for the soil moisture results.

**Sect.4:** We edited this section in part for better comprehensiveness and presentation. Per the editors' suggestion we tried to reduce any ambiguous and redundant writing.

**Figure 9:** We added a new figure showing water balance results from our simulations of drought.
* * *
**Reviewer 1**

We thank the reviewer for the careful review of this paper, and for the comments and suggestions provided. Below, we address the comments point by point. We are confident that all of the changes can be implemented, contributing to a stronger overall message and clearer communication of our key points.

1)      Delineation of "drought" and "non-drought" period is based on USDM data seems a bit random. 01-01-2008 to 31-12-2011 defined as "non-drought" period but it contains periods of "Extreme" and "Severe" droughts. Similarly, 01-01-2012 to 01-01-2019 "drought" period contains drought-free days along with period of "Extreme" and "Severe" droughts. Since this classification is a basis of the analysis that follows, a more robust classification, perhaps based on drought categories, is needed.

We acknowledge that our initial delineation of drought categories may seem arbitrary. In order to strengthen our analysis, we took the reviewer's suggestion of dividing the drought periods based on 3 categories that emerge directly from the Drought Monitor. Specifically, we have updated this by dividing our study period into three different categories. 1. no drought, 2. moderate drought (everything in categories D0 and D1) and 3. extreme drought (everything in categories D2–D4). We believe this

delineation better highlights the changes in onset and propagation of drought and provides a more robust characterization of the drought responses. We have added description of this step in the methods and updated all Figures where we show data in drought categories (i.e. Fig. 4, Fig. 5a, b, Fig 6a, b, c, Fig. 7).

2)      Analyzing and comparing PET and P between drought and non-drought periods, based on NMDC data seems like going in circles since NMDC drought categories are derived from the very dataset.

We would like to note that the data used in this study is from local weather stations at our two sites. We do not use the same data the NDMC uses. Furthermore, we would like to point out that the U.S. Drought Monitor bases its drought intensity categories on a wide range of indicators, including, but not exclusive to P and PET. We only use the Drought Monitor maps in this study to delineate periods falling into particular drought categories for our study area.

3)      NDVI derived from Landsat-5, Landsat-7, and Landsat-7 are not comparable and must be homogenized and filtered from clouds and other types of data noise (Goulden and Bales, 2019). I was unable to figure out if homogenization and cloud correction was performed or not. Also, considering the short growing season, a median NDVI value may not be appropriate as it may end up representing the NDVI at the beginning or end of the month. See Roche et al. 2018 for centering technique.

We really appreciate the thoroughness of the reviewer here. First, we only used cloudless images in our analysis. Second, based on the approach described in Goulden and Bales, 2019, we have homogenized our NDVI data and replotted relevant figures. We have also used the suggested centering method presented in Roche et al. (2018). These points have been updated in the methods.

4)      NDVI exhibits saturation beyond some threshold precipitation or available water, it can be seen in Figure 7a. You don't expect the NDVI to continue to increase with increasing water availability. Some vegetation expansion is possible when ample water supply is available and other resources (energy, nutrient etc.) are not limited but eventually max out. Fitting an exponential model ignores this fact.

Yes, it is true the NDVI saturates for any particular vegetation type. Therefore, we have introduced a threshold value that represents maximum greenness based on all historic values over the study period. These site-specific thresholds were then applied to our exponential model to prevent NDVI values from saturating. We have redone our analysis, updated this point in the methods, and replotted relevant figures. These changes did not significantly affect the results.

5)      The definition of polygons with homogeneous vegetation and soil textural properties requires further explanation. Considering the fact that you have a mixture of vegetation at both sites, how did you define "homogenous?

We based our delineation of these polygons on observations, both from the field and from remote sensing. The polygons were drawn in a way that were restricted to a common vegetation (grass) cover from NDVI, excluding other types of vegetation (i.e., trees). The assumption of homogenous soil properties is also based on field data, which was obtained via soil plots when the sites were set up by UCSB as part of IDEAS project. These points have been clarified in the methods.

6)      The scenarios can be better described in the methods, I could not understand Scenario A and B until looking at figure 9. What is the meaning of the truncated rainy season and how annual P from the truncated months are redistributed? Also, these scenarios represent intense future drought as posted in the research question (iii), but the presentation of results and discussion comes out as typical climate change scenarios.

Our rationale for a truncated season was to explore the effects of an even earlier onset of the dry season then what has already occurred. In this experiment, we are trying to simulate specific scenarios of climate change that would lead to more intense drought conditions (as discussed in the text). The truncated rainy season scenario was designed to explore the effects of an earlier shift in the onset of the dry season than what occurred in the recent drought. Spring rains are important to soil moisture stores and seed germination in grassland ecosystems. If the onset of the dry season were to shift towards early

spring, soil moisture stores would be exhausted earlier, senescence and browning of vegetation would start earlier, and lead to conditions more conducive to wildfires for more extended periods of time.

In Scenario A we shortened the rainy season to occur between November – March and rain in other months was lost. In Scenario B the same applies but the rainfall recorded after March was redistributed between Nov-Mar, effectively increasing the intensity of the rain events, but keeping seasonal totals the same. We have updated the description of these scenarios in the text to make things clearer.

7)      Figure 9 is interesting but can be conceptually predicted without running a model. Perhaps these results can be analysed to better understand the onset and longevities of the drought. Something similar to 5a but for different scenarios.

We agree that increased soil moisture drying, and drought responses could be predicted from a conceptual model. However, quantifying these changes for a real environment is not possible without running a model. It was our intent to apply our model to illustrate and quantify the potential changes of earlier drought onset and in response to different scenarios that may not yet have occurred. To further emphasize this, we have added to Fig 9 and the relevant discussion the relative % changes of time below the browning threshold to highlight the impacts of the different scenarios.

Minor points:

1)      Suggesting removing the unnecessary background information from the methods, i.e., do we need introductory sentences like these "Soil moisture is essential for plant growth and -health and accordingly, there are strong seasonal responses of vegetation to temperature and precipitation (Coates et al., 2015; Roberts et al., 2010)" to describe the study sites?

We disagree here. The work cited was specifically looking at drought and soil moisture deficits from a remote sensing perspective, and it is relevant to the region. It serves as a key background for our study. We have modified the sentence to link our study area and remote sensing to this reference.

2)      Precipitation values reported on top of the page 7 don't match the 20% difference reported on top of page 17

We acknowledge this mistake in our calculations. The actual average difference of precipitation per water year between the two sites is about 10%. We have corrected this in the manuscript.

3)      You mentioned inland site is not used for grazing, how about the coastal site?

The coastal site is also not used for grazing. We have added clarification to the relevant passage at L139.

4)      Provide mean temperature for the two sites.

Mean temperatures for both sites have been added to the relevant passage in the text in L143.

5)      Table S1, note the data formatting issue

We assume the noted issue refers to the number formatting of the silt content in Table S1. The issue has been noted and was corrected.

6)      Shortwave and longwave radiation measurements: are these net radiations?

Yes, these are net radiations. The meteorological stations at the coastal site includes a four-channel net-radiometer measuring upwelling and downwelling longwave and shortwave radiation, while at the inland site is equipped with a one-channel net radiometer. We have clarified this point in the text.

7)      L155: "For each site, we extracted daily maximum daytime temperatures, humidity and precipitation totals and calculated monthly averages to define the meteorology of the drought"- not clear. Which variables are daily maximum and which ones are totals? What do you mean by the monthly average of precipitation total?

Due to the high resolution of the data set (15min), temperature and relative humidity were summarized to diurnal maximum daytime values. Precipitation was summarized to daily totals. The passage in the text at L155 has been clarified.

8)      PET calculation using the Penman-Monteith model need more information on how other inputs were derived i.e., conductance, ground heat flux etc.

Due to the comprehensive nature of the dataset, a wide range of variables, such as net radiation, soil temperatures and windspeed, was available. This allowed us to estimate inputs such as conductance and soil heat flux and use them in our Penman Monteith calculations. We acknowledge the lack of information provided on this approach, so we have modified the text to add additional information in the relevant section at L157.

9)      Stevens hydro probe, provide manufacturer and model

Soil moisture content was measured using in-situ probes (Stevens Hydro Probe II, Stevens Water Monitoring Systems Inc., Portland). We have provided additional information on the manufacturer and model of the in-situ probes in L163.

10)      L166: here you argue for using the degree of saturation but then end up comparing VMC in Figure 9. Relative saturation may have been more appropriate as it accounts for differences in residual WC between the two sites.

The historical soil moisture data is presented as % saturation to account for the difference in soil textural properties between the two sites. We have also now presented our model results as saturation to maintain consistency.

11)      Fig S1 SMD can be equal to RAW as stated in the text

SMD can indeed be equal to RAW. If the reviewer refers to the line RAW<SMD< TAW in the text box, we acknowledge the mistake in the formulation and have changed it to RAW ≤ SMD ≤ TAW.

12)      Showing Fc, Wp, RAW, and TAW in figure 3 is misleading. The picture depicts a soil profile and not a unit volume. In its current form, it looks like the Wp is always at the bottom of the root zone.

The drawing is based on the FAO conceptual model, which can be found in Allen et al., (1998). Wilting point is indicated there in a similar way towards the bottom of the volume. However, we acknowledge that the indication of parameters and processes in 3-D may lead to confusion as pointed out by the reviewer. We have therefore changed the figure into 2-D to represent the bucket approach in a simplistic way.

13)      Equation 1, I don't quite understand what minimum and maximum Kc & median minimum and maximum NDVI means. Are not you regressing the monthly Kc values against monthly NDVI values with the index I being the month 1 through 12?

After some deliberation, we decided to take a different approach on estimating kc via remote sensing to make it more robust. Specifically, we adopted in the approach of Glenn et al. (2011), in which the crop coefficient can be replaced by vegetation indices (such as NDVI) that reflect the actual growth stage of the plant at the time of measurement. No reference values of kc are needed in this case but a direct relationship between kc and NDVI can be used to estimate ET. The text has been updated to reflect this change. This new approach enables a more dynamic kc than the previous approach, making the subsequent analysis more realistic.

14)      P-PET is not really a net precipitation, it is closer to aridity P/PET?

We agree, so have changed the terminology to 'available P (aP) for infiltration' to avoid confusion. We have updated the methods to clarify this point.

15)      L320 2012-2019 drought is only relevant for southern California. Statewide the drought ended in 2016.

We have changed the text in L320 to specify Southern California, rather than the whole state.

16)      16) Fig. 8: At what depth these soil moisture measurements were made? Is the simulated VMC are for the same depth or integrated over the entire root zone?

The measurements were taken at several depths (15,20,50cm); however, we are only interested in the balance of shallow soil moisture as it captures the dynamics of precipitation and evapotranspiration, we

are interested in. We use the shallow soil moisture observations to calibrate our model so that we are able to capture the main processes. The simulated moisture content represents an integrated bucket over the root zone and is not a direct reproduction of observed values. This point has now been clarified in the text.

**Reviewer 2**

We thank the reviewer for their positive feedback and their comments on our manuscript. We believe these comments can be straightforwardly incorporated into our manuscript and will strengthen the message and key points of the paper. Below we address their comments point by point.

General comments:

Not sure the title makes complete sense as written. Taken in parts… Onset of drought into soil moisture responses? Consider reworking, with a focus on natural grasslands.

We take the point that the title could have been clearer. We have therefore adopted a new title: 'Drought onset and propagation into soil moisture and grassland vegetation responses during the 2012–2019 major drought in Southern California'.

In general, I would recommend reducing use of "we" and shifting to a more passive tone throughout the manuscript. Shifts the emphasis from what the authors did and thought to a more objective viewpoint.

We respectfully disagree with reviewer and find passive voice to be less engaging to modern readers. We decided to maintain the use of active voice.

Parts are overly wordy and could use some critical editing to pare down to what is really essential to move the key story forward. The first paragraph of the introduction, for example, discusses forest response to drought in depth before getting to grasslands, which is the focus of the paper. Revisit the wording of the first sentence – could read something like "resulting in substantial impacts to water resources and ecosystems. These impacts varied regionally, depending on climate, elevation and biome… For example, upland forests: [1 or 2 sentences]. The impacts of drought on California's grasslands have been less well studied…"

Fair point. We have adapted the first paragraph to highlight the key points and set up the rationale of our research.

Specific comments:

L158: Add citation for the Penman-Monteith equation.

L158: Added.

L165: Explain the choice of the shallowest soil moisture observations as reference in this study. Why not the 20 or 50 cm measurements, which may be more representative of the root zone?

L165: We decided to use the shallowest soil moisture as reference because we are specifically interested in the response and behavior of the shallow soil moisture balance which comprises the majority of the moisture availability to grasses. The shallow sensor is capturing the precipitation and ET dynamics we were investigating, and we used it to calibrate our model to capture the dominant processes.

The simulated moisture content represents an integrated bucket over the root zone and is therefore not an exact reproduction of shallow moisture observations. In general, bucket model results cannot be directly compared to point measurements at specific depths without calibration.

Fig. 2: Can the time markers in this figure be at 1-1-YYYY rather than 1-4-YYYY (is that April 1)? Seems a little cleaner:and less ambiguous re. date format. Inclusion of 2008 and 2009 in the non-drought period is problematic, since most of SB County was under moderate-severe drought in those years. Why not separate out truly non-drought years, defined as years with some prescribed fraction in drought-free conditions, even if they are not temporally contiguous?

Fig 2: We acknowledge the confusion over the time markers in the figure. We adopted the format of the time markers from the USDM and are in US date format, thus start on January 4. We have changed the date format to a more coherent format of 01-01-YYYY. The issue of our initial classification of drought and non-drought periods was also mentioned by Reviewer 1, and we have made the changes outlined in the responses to Reviewer 1. We believe this strengthens our analysis and better highlights our findings.

Sec 2.4: Discuss noise reduction and normalization applied to the NDVI data obtained from Landsat's 5, 7 and 8 to generate multi-year timeseries. Were TOA band values used, or surface reflectance?

Sect. 2.4: We have addressed this point in our comments to Reviewer 1 and have adapted the suggested approach from Goulden and Bales (2019) to homogenize the NDVI data of the different Landsat missions. We used NDVI images produced by the USGS from surface reflectance in our analysis.

Sec 2.5.1: The description of the Soil Moisture Balance Model could be shortened significantly.

Sect. 2.5.1: Agreed. Shortened.

L215: "…to estimate potential evapotranspiration (PET)…"

L215: Changed.

Table 1 caption: Remove first of two "used" in first sentence.

Table 1 caption: Caption amended.

L253: This sentence does not read well. Is there a missing comma after NDVI?

L253: We have reworded this section to make it more coherent and easily understandable.

Sec 2.5.4: Seems to be an inconsistency in stated calibration range between line 267 (2008-2019) and L 272 (2008-2014). Please clarify.

Sect. 2.5.4. We acknowledge the inconsistency in the presentation of our methods. Data from 2008-2014 was used for calibration and the data from 2014-2019 was only used to validate the model. We have amended the relevant section to make this clearer.

L339: It is surprising that there is no difference in precipitation between drought and non-drought years. Perhaps a more stringent separation of these years would yield greater difference?

L339: We agree that separating the data into more stringent drought categories highlights the climatic differences more clearly. Dividing the data into three drought categories shows the underlying trend of declining precipitation totals during the drought periods.

L366: What is meant by "aggressive strategy"? Maybe a use a different term.

L366: We meant to emphasize the extreme increase in greenness as seen in the NDVI signal following the rainy season, as opposed to a more gradual increase in greenness. We accept the suggestion and have reworded this sentence. "These differences in the seasonal variation of NDVI suggest a strategy of rapid green up after winter rains, …".

L 451-452: The difference in response isn't only due to soil texture, right? Difference in climate (aridity) also drove response.

L451-452: Yes, the difference in the drought response is not only due to soil texture, but also due to the local differences in climate. The interaction between soil and climate led to the differing response we highlighted. We have reworded this sentence to make this clearer. "The results from contrasting sites (coastal and inland) corroborate studies showing differential responses to the drought over short distances due to spatial variation in soil texture as well as local climate and aridity …".

---

## Author Response (AR2)

We thank the reviewers for reading our revised manuscript and appreciate the comments. Below we address both reviewers' comments in detail.

Line numbers refer to clean version of the manuscript (with line numbers from reviewers' comments in brackets).

**Reviewer 1:**

Since PET is a notional variable, different definitions can be found in the literature. The concepts of $ET_0$, PET and atmospheric demand are frequently used interchangeably as noted by the reviewer. In this study we computed $ET_0$ via the Penman-Monteith equation using a reference crop surface and meteorological data from the weather stations. Then $ET_0$ was used to calculate AET.

We acknowledge the confusion and have changed all references of PET to $ET_0$ in the manuscript.

Within the model actual crop evapotranspiration under unstressed conditions is estimated as: $AET = kc*ET_0$, while under stressed conditions AET is reduced by the coefficient ks, which is calculated within the model and related to water availability. We believe that it is important to keep Eq. 3 in the manuscript, but we refer to AET rather than PET, which supports a fuller explanation of the estimation of kc. We also updated the link in the reference list.

**Fig. 6c:** Added correlation coefficient ($R^2$) to the graph.

**Eq. 4:** Changed regression coefficients to $\alpha$ and $\beta$.

**Reviewer 2:**

**L103 (L228):** Sentence was changed according to suggestion from the reviewer.

**Fig 1:** Added photos of the two study sites.

**L164 (L380):** The suggested corrections were made.

**L186 (L404):** The suggested correction was made.

**L193 (L409):** The sentence was edited.

**L196 (L413):** We clarified our approach of removal of cloudy images in the text.

**L210 (L415):** The suggested changes were made.

**L237 (L460):** Suggested sentence was removed.

**L330 - Sect.3.1** Renamed to "Climatology of the Drought".

**L 333 (L685):** The suggested changes were made.

**Fig 6:** Added legend to Fig. 6d,e to reduce ambiguity of the plotted NDVI of the different drought periods. Because the statistical difference is not easily visible in the violin plot for the inland site, we denoted any significance with \*\*\* in the plot.